# Targeting the Melanocortin 1 Receptor in Melanoma: Biological Activity of α-MSH–Peptide Conjugates

**DOI:** 10.3390/ijms25021095

**Published:** 2024-01-16

**Authors:** Ildikó Szabó, Beáta Biri-Kovács, Balázs Vári, Ivan Ranđelović, Diána Vári-Mező, Éva Juhász, Gábor Halmos, Szilvia Bősze, József Tóvári, Gábor Mező

**Affiliations:** 1HUN-REN–ELTE Research Group of Peptide Chemistry, 1117 Budapest, Hungary; ildiko.szabo@ttk.elte.hu (I.S.); biri.beata@gmail.com (B.B.-K.); mezo.diana@ext.oncol.hu (D.V.-M.); szilvia.bosze@ttk.elte.hu (S.B.); 2MTA-TTK “Momentum” Peptide-Based Vaccines Research Group, Institute of Materials and Environmental Chemistry, HUN-REN Research Centre for Natural Sciences, 1117 Budapest, Hungary; 3National Tumor Biology Laboratory, Department of Experimental Pharmacology, National Institute of Oncology, 1122 Budapest, Hungary; vari.balazs@oncol.hu (B.V.); randelovic.ivan@oncol.hu (I.R.); tovari.jozsef@oncol.hu (J.T.); 4School of Ph.D. Studies, Doctoral School of Pathological Sciences, Semmelweis University, 1085 Budapest, Hungary; 5Department of Pediatrics, Faculty of Medicine, University of Debrecen, 4032 Debrecen, Hungary; juhasze@med.unideb.hu; 6Department of Biopharmacy, Faculty of Pharmacy, University of Debrecen, 4032 Debrecen, Hungary; halmos.gabor@pharm.unideb.hu; 7Institute of Chemistry, Eötvös Loránd University, 1117 Budapest, Hungary

**Keywords:** α-MSH, melanoma, peptide–drug conjugates, in vitro antiproliferative effect, in vivo antitumor activity

## Abstract

Malignant melanoma is one of the most aggressive and resistant tumor types, with high metastatic properties. Because of the lack of suitable chemotherapeutic agents for treatment, the 5-year survival rate of melanoma patients with regional and distant metastases is lower than 10%. Targeted tumor therapy that provides several promising results might be a good option for the treatment of malignant melanomas. Our goal was to develop novel melanoma-specific peptide–drug conjugates for targeted tumor therapy. Melanocortin-1-receptor (MC1R) is a cell surface receptor responsible for melanogenesis and it is overexpressed on the surface of melanoma cells, providing a good target. Its native ligand, α-MSH (α-melanocyte-stimulating hormone) peptide, or its derivatives, might be potential homing devices for this purpose. Therefore, we prepared three α-MSH derivative–daunomycin (Dau) conjugates and their in vitro and in vivo antitumor activities were compared. Dau has an autofluorescence property; therefore, it is suitable for preparing conjugates for in vitro (e.g., cellular uptake) and in vivo experiments. Dau was attached to the peptides via a non-cleavable oxime linkage that was applied efficiently in our previous experiments, resulting in conjugates with high tumor growth inhibition activity. The results indicated that the most promising conjugate was the compound in which Dau was connected to the side chain of Lys (Ac-SYSNleEHFRWGK(Dau=Aoa)PV-NH_2_). The highest cellular uptake by melanoma cells was demonstrated using the compound, with the highest tumor growth inhibition detected both on mouse (38.6% on B16) and human uveal melanoma (55% on OMC-1) cells. The effect of the compound was more pronounced than that of the free drug.

## 1. Introduction

Melanoma is one of the three main types of skin cancer. Although melanoma is much less common than other skin cancers (it accounts for ~21% of all skin cancer incidences) [1], its extremely aggressive behavior makes it the leading cause of death among tumors of the skin. Melanoma originates from the malignant transformation of melanocytes, which are the melanin-producing cells of the skin, hair, and eyes. It is formed either by dysfunction of dysplastic nevi or by a single melanocyte [2]. Melanocytes are located with keratinocytes in the basal layer of the epidermis and form a very stable population, as they proliferate extremely rarely under normal circumstances. The inner layer (dermis) of the skin (that involves hair roots, blood, lymph vessels, and nerves) also contains melanocytes, but they are a biologically different population compared to the ones located in the epidermis. Since its first recognition by Clark et al. [3], melanoma has been determined to be heterogeneous, comprising a population of melanocytes of different origins and differentiation stages (from undifferentiated cancer stem cells with self-renewal, high proliferation, and differentiation capacity to functional melanocytes). Therefore, each melanoma variant behaves differently and has a different prognosis, which implies that there is no uniform treatment that can be used effectively [3]. Furthermore, because of their location, melanocytes have a high potential to spread rapidly to other parts of the body by entering the lymphatic system and bloodstream, resulting in a high rate of metastasis.

Melanoma can be treated effectively with surgery and/or radiation in the early stages (Stage I and II). However, at later stages (Stage III to V), when the malignant melanoma has spread and the lymph nodes are also involved or in the case of uveal melanoma, these treatments are not suitable, and chemotherapy is mainly used. The applied drugs, like alkylating agents (dacarbazine, the first FDA-approved drug for the treatment of metastatic melanoma, temozolomide, nitrosoureas), DNA cross-linking platinum drugs, and microtubule-toxin agents (taxanes and vinca alkaloids), result in a lower than 20% overall response and a less than 10% complete response during the treatment of patients. In addition, the developed drug resistance and toxic side effects, because of the lack of drug molecules, decrease the activity of the chemotherapeutic agents [4]. Therefore, the 5-year survival rates for malignant melanoma are less than 10%. These factors urge researchers to develop new types of drugs and/or treatment strategies. One of these might be a targeted therapy to overcome these drawbacks. Targeted tumor therapy, based on the tumor-specific or surface overexpressed receptors on cancer cell homing devices (e.g., antibodies or peptides) can be attached to anticancer agents. One of the most promising tumor cell surface markers is melanocortin 1 receptor (MC1R), a G-protein coupled receptor that has a pivotal role in melanogenesis and skin pigmentation due to the binding of its ligand, α-Melanocyte Stimulating Hormone (α-MSH). While the MC1R expression level is much lower in healthy melanocytes, melanoma cells frequently overexpress it. Therefore, this makes MC1R a useful marker for malignant melanoma as well as a potential target for melanoma diagnosis and therapy [5,6,7,8,9].

The overexpression of MC1R on malignant melanomas suggests that peptides such as α-MSH (Ac-Ser-Tyr-Ser-Met-Glu-His-Phe-Arg-Trp-Gly-Lys-Pro-Val-NH_2_, an *N*-terminal acetylated and *C*-terminal amidated tridecapeptide) regulate skin pigmentation in most vertebrates. The core α-MSH sequence His^6^-Phe^7^-Arg^8^-Trp^9^, conserved in several species, is sufficient for receptor recognition [10], while the C-terminal fragment is not essential for receptor binding. The presence of α-MSH receptors (MC1R) on both murine and human melanoma cells [11] suggests that α-MSH analogs can be developed into targeted melanoma imaging or therapeutic agents [12,13]. In addition, the hormone-receptor complex is rapidly internalized, and the receptor undergoes recycling within a few minutes. Based on these facts, several α-MSH analogs, linear and cyclic as well, have been intensively developed and published. Among them, the substitution of Met^4^ with Nle^4^ and Phe^7^ with D-Phe^7^ has yielded one of the most effective α-MSH analogs with sub-nanomolar receptor-binding affinity and resistance to enzymatic degradation [14,15,16]. In cyclic versions, the Lys^11^ is often replaced by Arg, suggesting that the Lys side chain is not necessary for receptor-binding affinity [17,18]. The α-MSH radiolabeled analogs have been predominantly used as diagnostic tools in the field of melanoma research, but only a few studies have investigated their chemotherapeutic applications [12,13,15,17,19,20,21,22,23,24,25,26]. Melphalan (phenylalanine mustard: PAM) was attached either to the side chain of Lys in position 11 (because it is not part of the main receptor recognition site of α-MSH) or to the *N*-terminus; furthermore, the Arg^8^ was replaced by PAM. However, the low stability of melphalan prevent sits use in therapy. Daunomycin was attached randomly to the three conjugation sites (*N*-terminus and the side chains of Lys in position 6 and 17) of β-MSH [27].

Therefore, our goal was to study the suitability of α-MSH as homing peptide for targeted drug delivery. For this purpose, we investigated α-MSH–daunomycin (Dau) conjugates. Several previous papers indicated that oxime-linked peptide–Dau conjugates are good tools for the selection of appropriate homing peptides, and they might also be potential drug candidates [28,29,30,31,32]. The autofluorescence property of Dau helps researchers to follow the cellular uptake of the conjugates by flow cytometry and/or confocal microscopy. Confocal microscopy is also suitable for following the uptake mechanism. The oxime linkage is non-cleavable (although it slowly decomposes in acidic conditions, especially below a pH of 2), but it has been demonstrated that a small metabolite is released in lysosomal homogenates or in the presence of lysosomal enzyme Cathepsin B, which is overproduced in tumor cells. This small metabolite (e.g., Dau=Aoa-Aaa-OH or H-Lys(Dau=Aoa)-OH, where Aoa is aminooxyacetyl moiety and Aaa is the first amino acid functionalized with Aoa) can bind to DNA similarly but with less efficiency than the free Dau) [33]. Furthermore, several developed peptide–Dau conjugates showed significant antitumor effects both in vitro and in vivo [28,30,32,34,35,36]. These data suggest that the peptide–Dau conjugates are suitable for carrying out many in vitro and in vivo experiments without the use of additional fluorescent or isotope labelling. It worth mentioning that some non-covalent combinations of α-MSH and doxorubicin have been applied with success. In addition, a doxorubicin–GnRH hormone peptide conjugates showed a good inhibition effect on uveal melanoma [37]. Thus, the peptide conjugates of anthracyclines might be good candidates for targeted therapy for different type of tumors, including melanoma.

## 2. Results

Different α-MSH peptide derivatives were synthesized to produce peptide–drug conjugates. The *N*-terminal acetylated and *C*-terminal amidated native tridecapeptide (Ac-SYSMEHFRWGKPV-NH_2_,) was used as a control with a slight modification. To avoid the unwanted oxidation of methionine during the synthesis, it was replaced with norleucine in the original sequence (Ac-SYSNleEHFRWGKPV-NH_2_) which is allowed by the literature data [38,39,40,41]. The native sequence was also truncated by focusing on the central part of the peptide (NleEHfRWGK). In this truncated peptide, another modification was done to improve the receptor binding affinity. The central region of the peptide contains phenylalanine, which, according to the literature, can be replaced by its D-configuration counterpart [14,15,16].

### 2.1. Synthetic Procedures and Chemical Characterization

The α-MSH peptides- full-length as well as truncated ones- were synthetized using Fmoc/^t^Bu strategy. To ensure drug conjugation, the peptides have been endowed with an aminooxy functional group by incorporation of the aminooxyacetyl (Aoa) group at different sites on the peptides. The following oxime-linked Dau–α-MSH conjugates were prepared: the drug was attached to the *N*-terminus of peptide (**Conj1**), or to the side chain of lysine in the sequence. In the latter case, the *N-*terminal part was acetylated (**Conj2** and **Conj4**, native and truncated ones, respectively). In addition, a conjugate containing two copies of the drug molecule was also developed (**Conj3**) (Figure 1).

The analytical characterization of drug-containing conjugates is shown in Table 1 and Appendix A.

### 2.2. Biological Characterization of the Full-Length α-MSH Drug Conjugates

First, we decided to determine and compare the in vitro and in vivo biological activity of Dau conjugates with the full-length α-MSH. For this purpose, the in vitro cytostatic activity and cellular uptake profile of the conjugates (**Conj1**, **2**, and **3**) were investigated. Furthermore, the in vivo tumor growth inhibitory effect was also determined on mouse melanoma allograft and xenograft models.

#### 2.2.1. In Vitro Antiproliferative Activity of Full-Length α-MSH Drug Conjugates

In vitro studies of the antiproliferative effect showed that melanoma-targeting Dau-conjugates have significantly higher efficacy on B16 murine melanoma cell lines in comparison to human melanoma cell lines (A2058, M24, and WM983B). The IC_50_ values were detected between 2 and 2.9 µM on B16 cells, where the highest activity was detected in the case of conjugate **Conj3**, followed by **Conj2** and **Conj1** (Table 2). Moreover, the relative potencies of conjugates to free Dau were also calculated as independent values from cell lines. A higher value of relative potency indicated the elevated targeting capacity of the conjugate on particular cell lines. Considering relative potencies, the best targeting capacity was shown to be **Conj3** on all four cell lines, followed by **Conj2** and **Conj1** (Table 2). The highest antitumor activity of **Conj3** on cells can be explained by the presence of two Dau molecules in this conjugate in comparison with **Conj2** and **Conj1**, which contain only one drug molecule.

#### 2.2.2. In Vitro Flow Cytometry Evaluation of Full-Length α-MSH Drug Conjugates

To better understand the subcellular mechanism of the newly synthesized Dau–α-MSH conjugates, we decided to monitor their cellular uptake. This could be easily carried out because of the fluorescence propensity of Dau: its emission can be detected both by flow cytometry and fluorescence microscopy. For quantification of the cellular uptake, flow cytometry studies were performed. First, all conjugates (**Conj1**, **2**, and **3**) were monitored on A2058 cells at different concentrations (Figure 2). All conjugates were taken up in a concentration-dependent manner. **Conj2** and **Conj3** showed the highest intensity, but interestingly, no significantly higher fluorescence values were detected in the case of **Conj3**, which contains two daunomycin molecules. Therefore, we assumed that **Conj2** entered the cells the most efficiently.

When considering the in vitro data, a good correlation was observed between the internalization ability and the cytostatic efficacy of the conjugates. **Conj1**, with the lowest internalization ability, had the highest IC_50_ value (9.8 ± 5.4 µM), while **Conj2** and **Conj3**, which had similar cellular uptake abilities, also had similar IC_50_ values (3.2 ± 0.4 and 3.0 ± 0.8 µM, respectively). Based on the in vitro data, it was concluded that the attachment of the drug to the *N-*terminus of the α-MSH peptide (**Conj1**) was not preferred and neither of the two drug molecules (**Conj3**) enhanced the in vitro efficacy of the conjugate.

#### 2.2.3. In Vivo Antitumor Effect of Full-Length α-MSH Drug Conjugates on B16 Melanoma Model

Prior to the in vivo studies of the antitumor effect of conjugates, an in vivo acute toxicity study was performed on healthy mice with the representative conjugate **Conj2** to determine the treatment dose. To investigate toxic effects and the tolerance of the Dau conjugates we determined the animal weight, behavioral and appearance changes and liver weight of the animals. After 14 days, no significant changes were observed in bodyweight (Appendix A), liver mass, or in the general appearance and behavior of the mice. However, the biochemical reflection of the liver, through analyzing the level of circulating aminotransferases which can serve as markers of hepatocellular injury and provide useful information about the liver toxicity [42], were not analyzed, which limit the toxicity study of conjugates. Based on the results, it was concluded that conjugate **Conj2** does not show acute toxicity for the animals up to the dose of 25 mg Dau content/kg, and that antitumor activity of Dau–α-MSH conjugates can be further investigated on tumor-bearing mice.

The effects of the Dau–α-MSH conjugates compared to the free drug were determined on subcutaneous B16 murine melanoma-bearing mice as a preliminary in vivo model. Animal bodyweights in the control and treatment groups increased at the end of the experiment in comparison to the start. Increasing bodyweights of 9.4 and 7.6% were observed in the control and free Dau groups, respectively, while the **Conj1**, **Conj2**, and **Conj3** groups showed an increase of 12.9, 8.5 and 18.2%, respectively (Figure 3a, and Appendix A). During the experiment, one animal died in the control group, 3 animals died in the group treated with free Dau, and 2 animals died in each group treated with Dau–α-MSH conjugates. The antitumor effect of the melanoma-targeting Dau–α-MSH conjugates was evaluated by measuring both tumor volumes and tumor weights in each group. Based on the tumor volume data presented in mm^3^, it was observed at the end of the experiment that the most potent conjugate (**Conj2**) inhibited tumor growth by 37.8%, while **Conj1** inhibited it by 23.8% (Figure 3b, Appendix A). A slight numeric (4.9%) inhibition was detected in the case of the free Dau treatment, while surprisingly, the average tumor volume size in the **Conj3**-treated group was 12.3% higher compared to the control group. Setting arbitrarily all tumor volumes as 100% at the starting point of the treatment, and following their percentage growth, the highest tumor growth inhibition was shown in the **Conj2**-treated group (75.4%, which was more pronounced by this type of calculation) than in the **Conj3**-treated group (14.7% inhibition), while the tumor volume in **Conj1** and the free Dau groups increased by 9.7 and 58.8%, respectively, in comparison to the control group (Figure 3c, Appendix A). Additionally, the antitumor effect of the melanoma-targeting Dau–α-MSH conjugates was evaluated by measuring tumor weights in each group after termination of the experiment (Figure 3d, Appendix A). Based on tumor weight, we determined that **Conj2** showed the highest inhibition again, inhibiting tumor growth by 38.6% in comparison to the control group. Free Dau inhibited tumor growth by 17% and **Conj1** inhibited growth by 2.4%, while tumor weight in the **Conj3**-treated group was increased by 9.2% in comparison to the control.

To conclude, **Conj2** exhibited the highest antitumor effect, followed by **Conj1**, while the lowest effect was obtained by **Conj3,** even though it contained two drug molecules. **Conj2** showed higher antitumor activity compared to the free Dau.

Comparing the results obtained in vitro and in vivo, it can be concluded that the position of the drug has a strong influence on the biological activity of the conjugates. Furthermore, the acetylated *N-*terminus in native α-MSH is important, and modification is not allowed in this position. Based on these results, **Conj2,** as the leading compound, was further modified for the next experiments.

### 2.3. Biological Characterization of the Sequentially Optimized α-MSH Drug Conjugate, Conj4

Starting from the results obtained by preliminary studies, we optimized the sequence of native α-MSH, focusing on the central region of α-MSH, considering the literature and experimental data. In vitro and in vivo measurements were also performed for this newly synthesized and optimized sequence-containing conjugate. In these studies, daunomycin and **Conj2** were used as positive controls.

#### 2.3.1. Experimental Model Selection Based on MC1R Expression

Since MC1R is known to be overexpressed on the surface of cancer cells in a subset of melanoma patients [11], we determined melanoma cell lines with a high expression of MC1R. First, we investigated the mRNA expression of MC1R by real-time qPCR (Figure 4) on several cell lines, namely OCM-1, OCM-3, SK-MEL-202, WM983A, WM983B, and A2058. As healthy controls, we used the skin fibroblasts CCD986-Sk, as well as HUVECs. The cell lines exhibited different expressions of MC1R, showing the highest expression in the case of OCM-1. WM983A and OCM-3 also had a relatively high expression of MC1R, displaying elevations of MC1R ten times higher than those of the healthy controls. Metastatic cell lines, such as WM983B, SK-MEL-202, and A2058, did not show highly elevated levels of MC1R mRNA compared to healthy controls.

#### 2.3.2. In Vitro Cytostatic Effect of the **Conj4** Compared to the **Conj2**

The in vitro cytostatic effects of Dau–α-MSH conjugates were compared with the PrestoBlue assay, which has a higher specificity and efficacy that of the previously used MTT assay, using the same six human melanoma cell lines used in the real-time qPCR assay. According to our results, free daunomycin exhibited the lowest IC_50_ values, and therefore the highest cytostatic effect, due to the fact that free daunomycin passively diffuses into cells [43,44]. Our Dau–α-MSH conjugates showed similar trends in the effectiveness of the tested cell lines. **Conj2** and **Conj4** resulted in IC_50_ values in the low μM range, (Table 3). A calculation of the targeting indices (TI—Table 3) showed that compound **Conj2** had an over 2-times higher targeting efficiency in most cell lines compared to the A2058 cell line (Table 3). Interestingly, WM983A showed the lowest TI among the high-MC1R-expressing cell lines compared to A2058, and WM983B exhibited high TI regardless of the relatively low MC1R expression on the mRNA level. In the case of **Conj4**, the treatment resulted in the highest TIs with OCM-1 and WM983B cell lines.

#### 2.3.3. In Vitro Flow Cytometry and Confocal Microscopy Evaluation

To compare the internalization ability and the intracellular localization of **Conj4** and **Conj2**, flow cytometric and fluorescence microscopic measurements were carried out. First, both conjugates were monitored on the above-mentioned six melanoma cell lines at different concentrations (Figure 5a,b). A concentration-dependent cellular uptake profile was observed for both conjugates, but a significant difference in the fluorescence intensity of the conjugates was detected. **Conj2** had a significantly higher (4–5-times) fluorescence intensity compared to **Conj4** in all cell lines, except for A2058, where the uptake was lower, but comparable to the other cell lines. The highest uptake levels were detected in the cases of SK-MEL-202, WM983A, and OCM-1. We also monitored the subcellular localization of the internalized conjugates by confocal microscopy (Figure 5c,d). First, A2058 cells were used to image the uptake of the conjugates. We could detect the presence of the Dau–α-MSH conjugates in the cytoplasm, partly co-localizing with lysosomes, as well as in the nuclei. In the case of this cell line, **Conj4** demonstrated higher uptake compared to **Conj2**, and could be detected in a higher amount in the nuclei (that correlated with the flow cytometry data) (Figure 2). In contrast, in the case of OMC-1 cells that showed higher MC1R mRNA expression, most of the Dau (probably the active metabolite) could be detected at their target compartment (in the nuclei) both at 12.5 μM and 25 μM concentrations. In line with the flow cytometry data, the signal for **Conj2** was higher compared to **Conj4** (Figure 5d). It is worth mentioning that confocal microscopy images were taken to detect subcellular localization, and not for quantitative measurements. Hence, Dau signals were set during image processing to show similar signal strengths to be able to compare localization (at the same settings, **Conj2** was not visible when **Conj4** was set to the level seen in Figure 5d).

Considering the above-described results obtained by RT-qPCR, cellular uptake as well as antiproliferative assays, OCM-1, SK-MEL-202 and WM983A cells proved to be the most promising for the establishment of murine experimental in vivo models.

#### 2.3.4. In Vivo Experiments

Following the first in vivo experiment, the sequence of the native α-MSH was optimized by focusing on the central region of the peptide. The efficacy of the newly designed and synthesized conjugate **Conj4** was tested in vitro and further in an in vivo experiment to determine the in vivo antitumor efficacy of **Conj4** compared to **Conj2** and the free drug (Dau).

With the three best-performing cell lines, OCM-1, SK-MEL-202 and WM983A, in the in vitro tests, we decided to investigate the tumor establishment and growth abilities in a murine experimental model. Cells were inoculated subcutaneously, and tumor progression and general conditions of the mice were monitored. In the case of SK-MEL-202, and WM983A cell lines, a solid tumor was established approximately 12–13 days after inoculation; however, we reported poor tumor growth (Figure 6). On the other hand, OCM-1 cells were able to establish a solid tumor 8 days after injection, and we witnessed exponential tumor growth reaching approximately 2000 mm^3^ at day 40. Moreover, the tumor that was established using the OCM-1 cell line grew similarly, regardless of the number of cancer cells injected. Since the general fitness of mice decreased in a similar pattern in all models, we decided to choose the OCM-1 cell line to establish our murine model, so that only efficacious Dau–α-MSH conjugates were able to affect the tumor expansion.

When investigating the effect of Dau–α-MSH conjugates and the free drug on tumor growth in vivo, male NOD-SCID mice were treated three times with saline as a control treatment and with each of the compounds; however, the doses varied based on the maximum tolerated dose (MTD) that was identified prior to the experiment (Appendix A). Mice were treated with 1 mg/kg with free drug, 5 mg/kg with **Conj2,** and 10 mg/kg with **Conj4**. Dau–α-MSH-treated and free-drug-treated groups started to separate in tumor size from the control group at day 23. The general condition of free-drug-treated mice declined rather quickly, and on day 28, mice were terminated. The rest of the groups were terminated on day 33, since the animals, including **Conj2**-treated mice reached the cut-off value for weight loss (Figure 7a). At the beginning of the treatment, free daunomycin seemed able to keep up with the efficacy of the Dau-αMSH treatment; on the other hand, it resulted in high toxicity and lack of tumor inhibition compared to the controls (Figure 7b). On Day 33, the growth of the tumor of **Conj2**-treated mice was reduced to 55% (p: 0.0056) of the control size. Moreover, **Conj4** was also able to reduce the tumor size by 14% compared to the control; however, this did not seem to be a statistically significant inhibition (Figure 7b). Based on the liver weights, neither the **Conj2** and **Conj4** nor the daunomycin showed significant toxicity (89.93%, 94.65% and 89.79% compared to the control, respectively; Appendix A). This toxicity data correlated well with the loss of bodyweight in the animals (Figure 7a).

## 3. Discussion

Chemotherapy, which is one of the main approaches for cancer treatment, is mostly ineffective for advanced melanoma due to its high metastatic nature. The failure of this classical treatment requires the development of more effective therapeutic agents and treatment strategies. MC1R overexpression can be detected in over 80% of melanoma patients [11]. The natural ligand of MC1R is the α-MSH hormone peptide. Interestingly, α-MSH derivatives are very rarely used as homing devices in drug targeting. Several publications showed conjugates for tumor detection in which chelated radioligands were connected to the *N-*terminus of the peptide [12,19,22,45]. A few papers also described targeted nanoparticles with α-MSH as a targeting molecule. In these cases, the nanoparticle was also attached to the *N-*terminus of α-MSH derivatives [20,23,24]. Therefore, we decided to prepare α-MSH–drug conjugates that may help to determine a structure-activity relationship suitable for further development. It has been documented that various tumor cells produce different hormones and express specific receptors to recognize these hormone molecules. It serves as a basis for tumor cells to maintain their division through autocrine-paracrine regulation. This is also true and well established for melanoma cells, which can produce the hormone precursor (POMC) as well as POMC-derived peptide hormones, such as α-MSH. Since melanocytes expressed MC1R and POMC—which can affect also endogenously MC1R and other MC receptors—rational combination of targeting peptides of MSH and POMC peptides can provide much higher inhibition of the tumor [46,47]. Among peptide hormones, agonistic analogs of GnRH and somatostatin can inhibit tumor growth [48], while in other cases, such as with α -MSH, only antagonists can evoke tumor growth inhibition in an autocrine/paracrine manner. Consequently, agonist analogs can stimulate tumor growth in a similar way to the native endogenous hormone. However, the drug-containing conjugates can be specifically taken up by tumor cells and through their metabolism the released drug or its active metabolites are able to eliminate tumor cells.

As a first step, we planned to study the influence of the conjugation site and length of the homing peptide on the biological activities. It was indicated in our recent study [31] that the conjugation site has a significant influence on the antitumor activity of conjugates because it affects receptor binding, cellular uptake, and drug release. In addition, cost-effectiveness in drug development is a key factor; therefore, shorter homing peptides that can maintain their effectivity might be better.

In this study, the native sequence of α-MSH was used, in which Met was replaced by Nle (SYSNleEHFRWGKPV). Daunomycin was attached to the amino functional groups via non-cleavable oxime linkage. These types of conjugates have many excellent properties to study and compare the bioactivity of the compounds. One of the developed conjugates contained drug molecules on both conjugation site (*N-*terminal and the side chain of Lys). The two additional conjugates were modified by the drug, either on the *N-*terminus or on the Lys side chain. The in vitro cytostatic effect on mouse melanoma cells did not show any significant differences among them. However, results indicated that the conjugates with Dau on the side chain of Lys could enter the cells more rapidly and efficiently. In contrast, the in vivo tumor growth inhibition was the most pronounced in the case of Ac-SYSNleEHFRWGK(Dau=Aoa)PV-NH_2_ (**Conj2**). It is worth mentioning that the dose was normalized to Dau content, thus the injected dose of **Conj3** was half that of **Conj 2**. Nevertheless, **Conj2**, with one molecule of the drug, might be superior to the conjugate containing two molecules of daunomycin. The higher tumor growth inhibition effect of **Conj2** over **Conj1** where the Dau is connected to the *N-*terminus, confirms our previous results with Dau-GnRH-III (Glp-His-Trp-Lys(Bu)-His-Asp-Trp-Lys(Dau=Aoa)-Pro-Gly-NH_2_), suggesting that H-Lys(Dau=Aoa)-OH can release easily from this position by the dipeptidyl-peptidase activity of cathepsin B.

In the second experiment, our goal was to identify whether the truncation of the terminal parts of the homing peptide (in **Conj2**)—but at the same time keeping the central sequence (**Conj4**) that has responsibility in receptor binding—was allowed without the loss of biological activity. In **Conj4**, the Phe was replaced by its D-isomers, as these have been suggested to increase the receptor-binding affinity of α-MSH derivatives [14,15,16]. In spite of this observation, the conjugate with the truncated peptide showed significantly lower in vitro cytostatic effect, cellular uptake on human melanoma cells, and in vivo tumor growth inhibition on human OCM-1 melanoma. The results suggested that the significant shortening of the peptide sequence is not allowed. It was also concluded that uveal melanoma OCM-1 is a good human melanoma experimental model because of its high MC1R expression and fairly good growing capability in comparison with other human melanoma models. It is worth noting that immune cells also express MC1R [49], and the effect of the conjugates may be influenced by their interaction with immune cells. However, in two in vivo models—an immunocompetent and an immunodeficient mouse model—we did not observe any significant difference in their bioactivity.

## 4. Materials and Methods

### 4.1. Materials

The materials applied for the synthesis and conjugation of peptides, with their abbreviations, are listed in the Appendix A.

### 4.2. Synthetic Procedures and Chemical Characterization

The α-MSH peptide derivatives (native and truncated ones) were synthesized manually on Rink Amide MBHA resin (0.67 mmol/g capacity) using a standard Fmoc/^t^Bu strategy with DIC-HOBt coupling reagents. However, when Dau was conjugated to the side chain of Lys in position 11, the ε-amino group was protected with a selectively removable Dde protecting group. At the end of the synthesis of the peptide sequence, the *N-*terminal Fmoc group was cleaved with 2% piperidine + 2% DBU in DMF (in four steps 2 + 2 + 5 + 10 min); then, they were handled differently depending on the types of conjugates. In the case of the synthesis of **Conj2** and **Conj4**, after the *N-*terminal acetylation (Ac_2_O:DIEA:DMF (1:1:3, *v*/*v*/*v*%) the Lys side chain protecting group (Dde) was selectively cleaved with 2% hydrazine hydrate (6 × 2 min). On the other hand, for the synthesis of two Dau-containing conjugates (**Conj3**), both the Fmoc and the Dde protecting groups were removed, respectively. Boc-Aoa-OH was attached either to the *N-*terminus and/or to the ε-amino group of the Lys still on the resin using the standard coupling agents.

Peptides were cleaved from the resin with TFA/H_2_O/TIS (9.5:2.5:2.5, *v*/*v*/*v*) mixture (2 h, RT). After filtration, compounds were precipitated in cold diethyl ether, centrifuged (4.000 rpm, 5 min), and freeze-dried from water. The crude peptides were characterized before the ligation by analytical RP-HPLC and ESI-HRMS. The crude compound was clean enough and used in the next synthetic step without any purification.

Oxime ligation was performed as described previously [33]. Briefly, the aminooxy-functionalized α-MSH peptides were dissolved in 0.2 M NaOAc solution (pH 5.2) and Dau·HCl (10% excess to the peptide) was added to the solution. Oxime ligation was carried out almost quantitatively in a day. The oxime bond-linked Dau–α-MSH conjugates were characterized by analytical RP-HPLC and ESI-HRMS (Appendix A).

### 4.3. Reverse Phase High-Performance Liquid Chomatography (RP-HPLC)

The purification of crude peptides and conjugates was performed on an UltiMate 3000 Semiprep HPLC (Thermo Fisher Scientific, Waltham, MA, USA) with a Phenomenex Jupiter Proteo C-12 column (250 × 10 mm) using gradient elution, consisting of 0.1% TFA in water (eluent A) and 0.1% TFA in acetonitrile/water = 80/20 (*v*/*v*) (eluent B).

The crude and purified peptides were analyzed by analytical RP-HPLC (Shimadzu prominence HPLC system) with a Phenomenex Jupiter Proteo C-12 column (150 × 4.6 mm) using gradient elution, consisting of 0.1% TFA in water (eluent A) and 0.1% TFA in acetonitrile/water = 80/20 (*v*/*v*) (eluent B).

### 4.4. Electrospray Ionization-High-Resolution Mass Spectromerty (ESI-HRMS)

The identification of the products was achieved by high-resolution mass spectrometry using a Thermo Scientific Q Exactive Focus Hybrid Quadrupole-Orbitrap Mass Spectrometer. Samples were dissolved in 50% acetonitrile–50% water containing 0.1% formic acid. Mass spectra were recorded in positive mode in the m/z 200–1500 range.

### 4.5. Cell Culturing

The following cell cultures were used for the biological evaluation of the synthetic melanoma-specific peptide–drug (Dau–α-MSH) conjugates: A2058 (human skin melanoma, CVCL_1059), M24 (human metastatic skin melanoma, CVCL_D032), OCM-1 (human ocular choroidal melanoma-1, CVCL_6934), OCM-3 (human ocular choroidal melanoma-3, CVCL_6937), WM983A (human metastatic skin melanoma, CVCL_6808), WM983B (human metastatic skin melanoma CVCL_6809), SK-MEL-202 (human skin melanoma, ATCC HTB-68) and B16 (mouse melanoma, ATCC: CRL-6475). Cells were maintained in RPMI-1640 (Lonza, Basel, Switzerland), supplied with 10% FBS (fetal bovine serum; BioSera, Nuaille, France), L-glutamine, and 1% penicillin–streptomycin (from 10 000 units penicillin and 10 mg streptomycin/mL, Gibco, Dublin, Ireland) at 37 °C in a 5% CO_2_ atmosphere. No mycoplasma contamination was detected in the cell cultures.

### 4.6. Determination of mRNA Expression Level of Melanoma Cells by qPCR

The relative RNA expression of MC1R gene was determined using the RT-qPCR method. The total RNA from the investigated cell lines was isolated using Trizol^®^ reagent (Ambion, by Life Technologies, Carlsbad, CA, USA). The concentration and the quality of the RNA samples obtained were measured using a spectrophotometer (NanoDrop ND-1000, Wilmington, DE, USA) at an absorbance of 260 nm and 280 nm. The cDNA synthesis was completed in Eppendorf 5331 Mastercycler Gradient thermocycler (Eppendorf, Enfield, CT, USA). A total of 500 ng of total RNA was transcribed into cDNA according to the protocol of the Reverse Transcription System provided by Promega (Promega, Madison, WI, USA). Afterwards, the cDNA samples were stored at −20 °C until further processing. The primers were obtained from Sigma-Aldrich, St. Louis, MO, USA, and were designed based on the reference sequence obtained from NCBI RefSeq (NM_002386.4). Primer sequences are as follows: MC1R_forward-CATCGCCGTGGACCGCTACATC, MC1R_reverse-GCTGAAGACGACACTGGCCACC. Relative expression of MC1R was measured using SsoAdvanced Universal SYBR^®^ Green Supermix assay (Bio-Rad, Hercules, CA, USA) with a CFX96 Touch Real-Time PCR Detection System (Bio-Rad, Hercules, CA, USA). Relative expression was determined by normalizing the expression levels to human β-actin mRNA levels as housekeeping genes and by comparing the MC1R mRNA levels of the investigated cell lines to the healthy skin fibroblast cell line, CCD-986-Sk. To estimate the relative expression, the delta-delta Ct method was used. The data represent three independent experiments, each performed in 3 technical replicates.

### 4.7. Determination of the In Vitro Antiproliferative Activity

The in vitro cytostatic effect of Dau–α-MSH conjugates was determined by two different assays. The full-length Dau–α-MSH conjugates were investigated by MTT assay [50,51,52,53,54] using human and murine melanoma cell cultures. For determination of the cytostatic effect of these conjugates, the cells were treated with the compounds at 0.8–100 μM concentration range dissolved in the corresponding serum-free media for 24 h. After incubation, the treating solutions were removed, the cells were washed two times with serum-free medium and cultured for 48 h in the appropriate serum-containing cell culture medium. On the fourth day, the MTT assay was carried out to determine the IC_50_ values of the compounds. IC_50_ is the concentration that inhibits cell proliferation by 50%. Nonlinear regression analysis was performed with Prism 6 software (GraphPad, La Jolla, CA, USA) to generate sigmoidal dose–response curves, from which the 50% inhibitory concentration (IC_50_) values of the compounds were calculated and presented as micromolar (µM) units. The experiments were done in triplicate, and each experiment was repeated twice.

The in vitro cytostatic effect of the Dau–α-MSH conjugate with optimized sequence (**Conj4**) was investigated and compared with **Conj2** by PrestoBlue assay [55,56,57] using human melanoma cell cultures with different origins. For determination of the cytostatic effect of the conjugates, the cells were treated with the compounds at 0.003–200 μM concentration range dissolved in the corresponding serum-free medium for 72 h. After the treatment, the PrestoBlue assay (purchased from Invitrogen (Waltham, MA, USA) was carried out to determine the IC_50_ values of the compounds. Based on the IC_50_ values, we also determined the targeting index (TI) of the conjugates using the following equation [58]:$$TI=\frac{\left(\frac{IC50neg}{IC50pos}\right)conjugate}{\left(\frac{IC50neg}{IC50pos}\right)free\;drug}$$

### 4.8. In Vitro Flow Cytometry Evaluation

The human melanoma cell cultures with different origins were cultured as described above. To study the cellular uptake of the Dau–α-MSH conjugates, 10^5^ cells per well were plated on 24-well plates one day prior to the experiment. After 24 h incubation at 37 °C, cells were treated for 3 h with the compounds solved in the corresponding serum-free medium. The cellular uptake of the compounds was investigated in the 12.5–50 µM concentration range. Cells treated with serum-free medium for 3 h were used as a control. After incubation, the medium was removed, and the cells were treated with 100 µL trypsin for 10 min. Trypsin digestion was stopped by the addition of 900 µL HPMI medium (9 mM glucose, 10 mM NaHCO_3_, 119 mM NaCl, 9 mM (4-(2-hydroxyethyl)-1-piperazineethanesulfonic acid (HEPES), 5 mM KCl, 0.85 mM MgCl_2_, 0.053 mM CaCl_2_, 5 mM Na_2_HPO_4_ × 2H_2_O, and pH 7.4) containing 10% FBS, and the cells were moved from the plate to FACS tubes. Cells were centrifuged at 216 g at 4 °C for 5 min and the supernatant was removed. After this procedure, cells were resuspended in 250 µL HPMI, and the increase in the fluorescence intensity of different types of melanoma cells was monitored by flow cytometry (BD LSR II, BD Bioscience, San Jose, CA, USA). Data were analyzed with FACSDiVa 5.0 software.

### 4.9. Immunostaining and Confocal Microscopy

A2058 and OCM-1 cells were seeded to coverslip-containing (thickness 1, Assistant, Karl Hecht GmbH & Co. KG, Sondheim/Rhön, Germany) 24-well plates (Sarstedt, Nümbrecht, Germany) one day prior to treatment at a density of 5·× 10^4^ cells/well. Cells were treated with 12.5 and 25 μM of Dau–α-MSH conjugates that were diluted in a serum-free medium for 3 h. Lysosomes were stained with LysoTracker^TM^ Deep Red (Thermo Fisher Scientific, Waltham, MA, USA, 300 nM for 30 min, followed by nucleus staining with Hoechst 33342 (0.2 μg/mL, 10 min). After washing with phosphate-buffered saline (PBS, Lonza, Basel, Switzerland), cells were fixed with 4% paraformaldehyde for 20 min at 37 °C. Coverslips were mounted to microscopy slides by Mowiol^®^ 4–88 mounting medium (Sigma-Aldrich, St. Louis, MI, USA). Confocal microscopy images were acquired on a Zeiss LSM 710 (in case of A2058 cells) or Zeiss LSM 780 confocal microscope (in case of OCM-1 cells) (Carl Zeiss Microscopy GmbH, Jena, Germany) using a Plan-Apochromat 40×/1.4 Oil DIC M27 objective. Hoechst 33342 and daunomycin-conjugates and LysoTracker Deep Red were excited with lasers 405, 488 and 633 nm, respectively. ZEN Lite (Carl Zeiss Microscopy GmbH, Jena, Germany) software was used for image processing.

### 4.10. Experimental Animals

Different murine models were used for in vivo experiments. For determination of in vivo toxicity as well as for in vivo antitumor activity of the drug and the conjugates, adult male BALB/c, C57BL/6 or NOD-SCID mice were used. Mice were kept in a sterile environment in Makrolon^®^ cages at 22–24 °C (40–50% humidity), with light regulation of 12/12 h light/dark. The animals had free access to sterilized tap water and were fed a sterilized standard diet (VRF1, autoclavable, Akronom Kft., Budapest, Hungary) *ad libitum*. Animals used in our study were taken care of according to the “Guiding Principles for the Care and Use of Animals” based on the Helsinki declaration, and the study was approved by the ethical committee of the National Institute of Oncology. Animal housing density was in accordance with the regulations and recommendations from directive 2010/63/EU of the European Parliament and of the Council of the European Union on the protection of animals used for scientific purposes. Permission license for breeding and performing experiments with laboratory animals: PEI/001/1738-3/2015 and PE/EA/1461-7/2020.

### 4.11. Acute Toxicity Study of Drug, Conj2 and Conj4

To determine toxicity of conjugates on healthy animals, in vivo acute toxicity study of Dau, **Conj2** and **Conj4** was performed. In the first experiment, adult BALB/c male mice (29–31 g) were exposed by one injection of the conjugate at the start of experiment, by intraperitoneal (i.p.) administration. A dose of 25 mg Dau content/kg was used, in a volume of 0.1 mL per mice (3 mice per condition). In the second in vivo experiment, NOD-SCID mice were exposed by four injections of the conjugates at the start of experiment, by intraperitoneal (i.p.) administration. Doses of 5, 10 and 15 mg Dau content/kg were used for the toxicity assay, at a volume of 0.1 mL per mice (3 mice per condition). The animals were kept under the conditions as described above.

### 4.12. In Vivo Antitumor Effect of Drug, Conj2 and Conj4

In the first experiment, B16 murine melanoma cells were subcutaneously (s.c.) injected into side of the latero-abdominal region of 35 C57BL/6 male mice (20–28 g), which murine strain is syngeneic for the B16 melanoma tumor cell line [59,60], 6.5 ×·10^5^ cells in a volume of 200 µL M199 medium was inoculated per animal. Mice were randomized in a stratified procedure considering two main factors. The primary factor was to obtain groups with the most equal tumor sizes, the secondary factor was to have groups in which the weight of mice are the most equal. The treatment started 9 days after cell inoculation when average tumor volume was 30 mm^3^. Compounds were dissolved in saline solution (Teva, Debrecen, Hungary), and administered via i.p. injection in a volume of 0.1 mL per 10 g of bodyweight. 5 groups by 7 animals were established and treated with the following doses and schedule: the mice in the control group were treated with the solvent; free daunomycin (Dau) treated group (1 mg/kg, treatments on days 9 and 17); groups treated with **Conj1**, **Conj2,** and **Conj3** (10 mg/kg Dau content, treatments on days 9, 13, 15, 17 after cell inoculation). Animal weight and tumor volumes were measured initially when the treatment started and at periodic intervals following treatment. A digital caliper was used to measure the longest (a) and the shortest diameter (b) of a given tumor. The tumor volume was calculated using the formula V = ab^2^ × π/6, whereby a and b represent the measured parameters (length and width). Termination of the experiment was 20 days after cell inoculation, i.e., 12 days after treatment started, since the average volume of the tumors in the control group reached over 1800 mm^3^. The mice from all groups were sacrificed by cervical dislocation after which their tumors were harvested and weighed for antitumor effect assessment. Antitumor effect of treatments was evaluated measuring the tumor volume and calculating the percentage of how much the tumor volume grew in comparison to the starting tumor volume which was set arbitrarily for all tumors as 100% at the start of treatment.

In the next experiment, the murine xenograft melanoma model establishment studies were performed using 8–12-week-old male NOD-SCID mice. For selection of the most suitable tumor model, three cell lines were then inoculated subcutaneously, OCM-1, SK-MEL-202, and WM983A. Cells were injected in a 0.1 mL RPMI-1640 basic medium in a concentration of 1.5·× 10^6^ cells/mL, and the tumor growth and weight of mice were monitored 2–3 times per week. Subsequently, the antitumor effects of the Dau–α-MSH conjugates were determined in a mouse xenograft melanoma model selected as described above. Once the tumor volume reached 60 mm^3^, mice were randomized and assigned to different groups for each treatment: 0.9% saline as a control, free daunomycin, **Conj2** and **Conj4**. Treatments were injected intraperitoneally every fifth day, three times in total (daunomycin dosage of 5 mg/kg in case of **Conj2**, 10 mg/kg for **Conj4**, respectively, which was determined based on the toxicity tests, and 1 mg/kg were used for free drug–MTD). The weight and tumor size of mice were monitored during the whole experiment. On Day 33, mice were euthanized, and the primary tumor, heart, lung, liver, and spleen were harvested and stored in 4% formalin (Molar Chemicals, Halásztelek, Hungary). Liver weight was measured for the calculation of the toxicity of conjugates and daunomycin.

### 4.13. Statistical Analysis

In the case of the in vivo allograft model, statistical analyses were performed by GraphPad Prism 6 (GraphPad Software, San Diego, CA, USA) using the non-parametric Mann–Whitney test, where *p*-values lower or equal than 0.05 were considered statistically significant.

## 5. Conclusions

In conclusion, we have found that the α-MSH–drug conjugate system is suitable for comparing different peptide–drug conjugates and selecting the most promising structure. The best conjugate showed moderate but better in vivo antitumor activity than the free drug. Thus, with further modifications (e.g., substitutions to increase efficacy and stability, incorporation of linkers and drug molecules), it may be suitable for the preparation of potential chemotherapeutic agents for targeted therapy of melanoma.

## Figures and Tables

**Figure 1 ijms-25-01095-f001:**
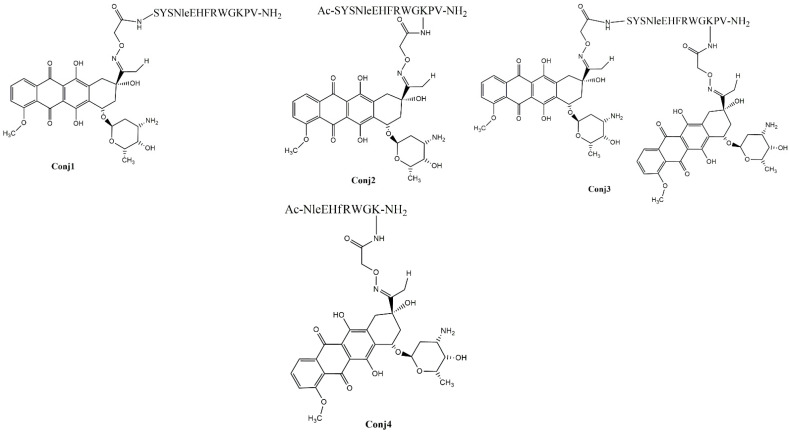
Chemical structure of the Dau–α-MSH conjugates.

**Figure 2 ijms-25-01095-f002:**
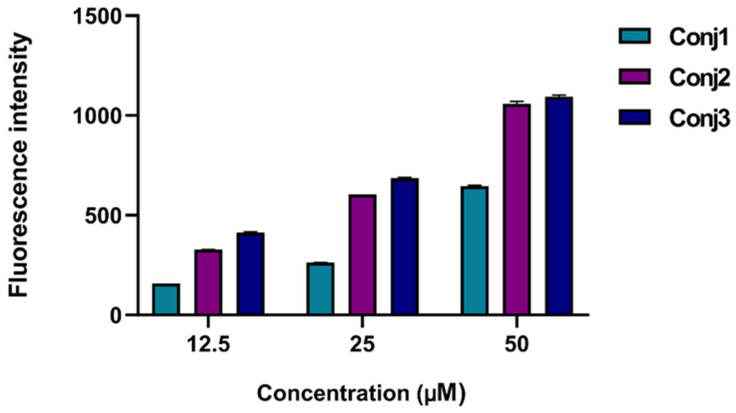
Determination of cellular uptake of Dau–α-MSH conjugates measured by flow cytometry. A2058 cells were treated with the conjugates in a 12.5–50 µM concentration range for 3 h. The increase of the fluorescence intensity of melanoma cells was monitored by flow cytometry (BD LSR II, BD Bioscience, San Jose, CA, USA). Data were analyzed with FACSDiVa 5.0 software.

**Figure 3 ijms-25-01095-f003:**
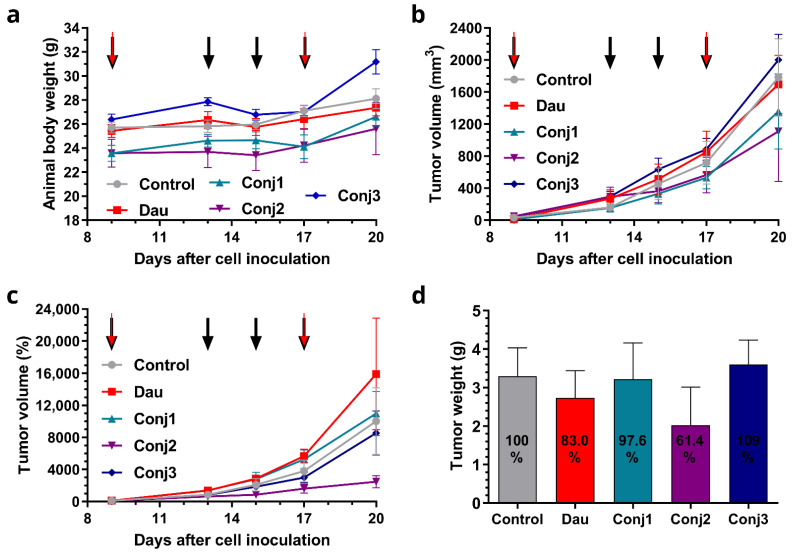
Effect of Dau–α-MSH conjugates (10 mg Dau content/kg, black arrows) and free Dau (1 mg /kg, red arrows) in subcutaneous B16 murine melanoma bearing C57BL/6 male mice in vivo: (**a**) animal bodyweight (grams, average ± SEM); (**b**) tumor volume (mm^3^, average ± SEM); (**c**) tumor volume (percentage, average ± SEM); and (**d**) tumor weight (grams, average ± SEM). Seven animals per group.

**Figure 4 ijms-25-01095-f004:**
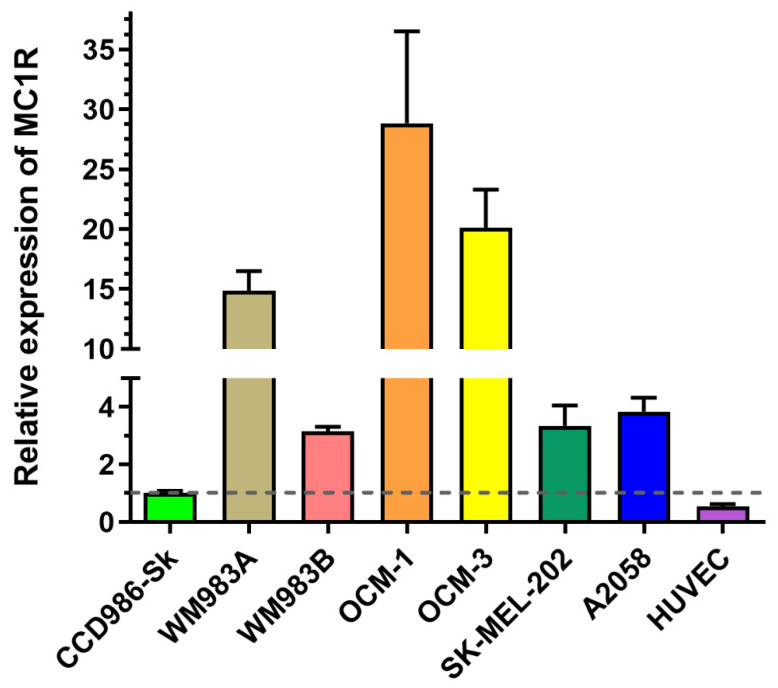
Determination and comparison of mRNA expression level of MC1R by real-time qPCR on different melanoma and healthy control cells. Relative expression was determined by normalizing the expression levels to the healthy skin fibroblast cell line, CCD-986-Sk (black dashed line). The data represent three independent experiments, each performed in 3 technical replicates.

**Figure 5 ijms-25-01095-f005:**
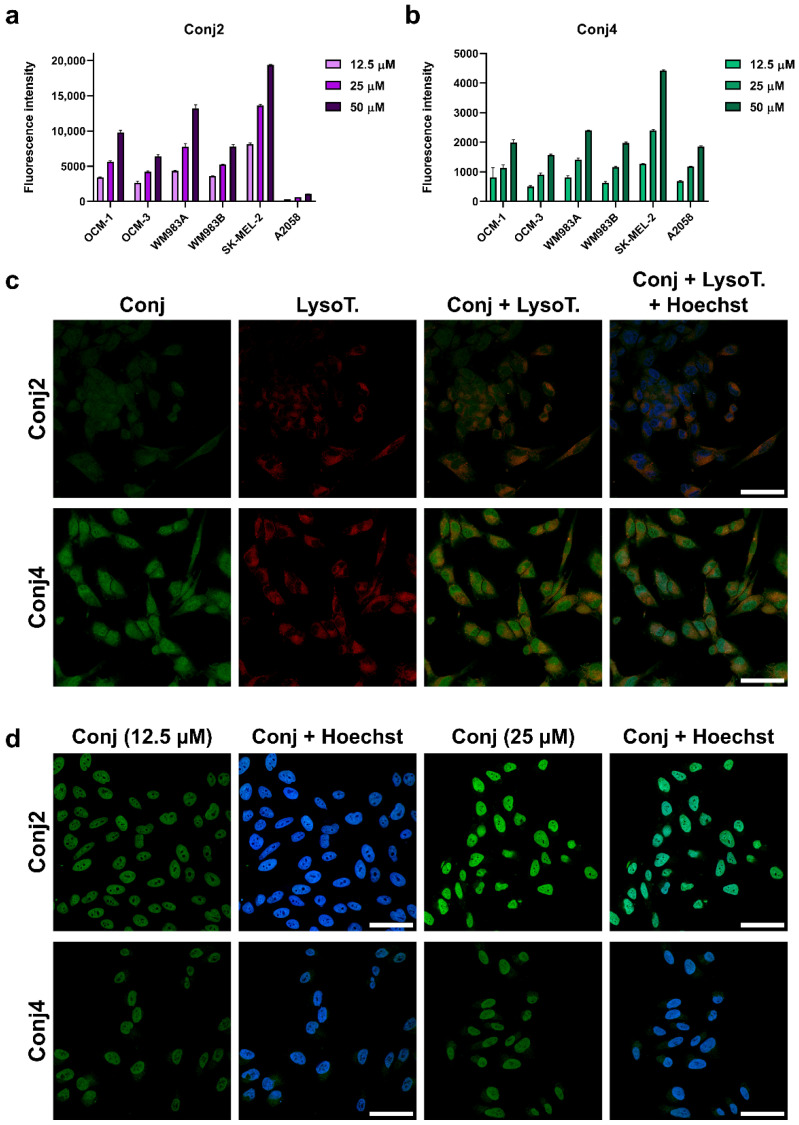
Comparison of cellular uptake profile and subcellular localization of **Conj2** and **Conj4** measured by flow cytometry and confocal microscopy: comparison of the uptake of **Conj2** (**a**); and **Conj4** (**b**) on different melanoma cells after 3 h by flow cytometry. Subcellular localization of Dau-αMSH imaged by confocal microscopy: (**c**) comparison of uptake of **Conj2** and **Conj4** (green) by A2058 cells after 3 h incubation. Lysosomes were stained with LysoTrackerTM Deep Red (red), nuclei were stained with Hoechst 33342 (blue); and (**d**) localization of uptake of **Conj2** and **Conj4** by OCM-1 cells. Scale bars represent 50 μm.

**Figure 6 ijms-25-01095-f006:**
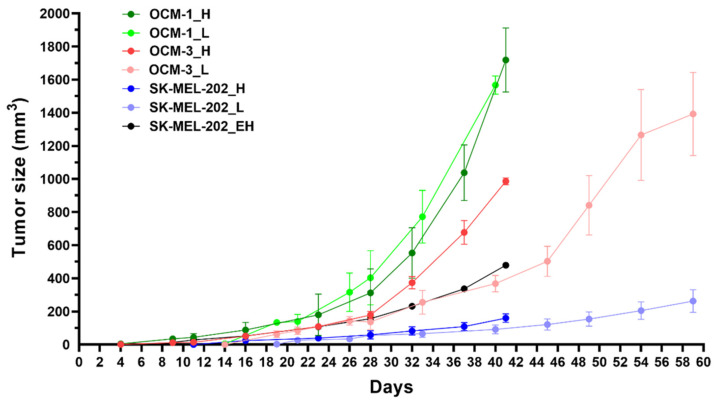
Determination of tumor establishment and growth abilities of the most promising cell lines (OCM-1, SK-MEL-202 and WM983A) in murine experimental model. L: low inoculated cell number (500,000 cells/mice), H: high inoculated cell number (1,000,000 cells/mice), EH: 2,000,000 cells/mice.

**Figure 7 ijms-25-01095-f007:**
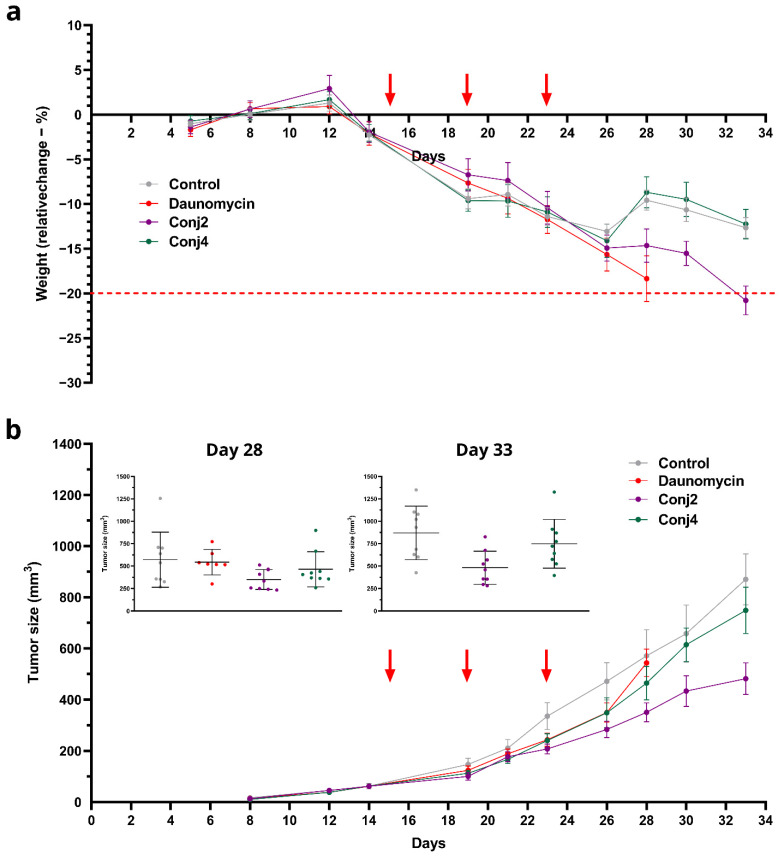
Determination of in vivo tumor growth inhibitory effect of **Conj2** and **Conj4** on OCM-1 tumor bearing murine model: (**a**) bodyweight of the mice (grams, average ± SEM), the red dashed line represents the critical weight loss of the animals as proposed in the EU Directive standards and recommendations.; (**b**) Tumor volume (mm^3^, average ± SEM). 7 animals per group. The red arrows on the figures show the days of treatments (days 15, 19 and 23).

**Table 1 ijms-25-01095-t001:** Chemical characterization of Dau–α-MSH peptide conjugates.

Code	Sequence	tR (Min) ^1^	Mmo (Da) ^2^		
			Calc	Meas	Differences (ppm)
**Conj1**	Dau=Aoa-SYSNleEHFRWGKPV-NH_2_	12.6	2187.0266	2187.0080	18.56
**Conj2**	Ac-SYSNleEHFRWGK(Dau=Aoa)PV-NH_2_	13.0	2229.0371	2229.0214	15.66
**Conj3**	Dau=Aoa-SYSNleEHFRWGK(Dau=Aoa)PV-NH_2_	12.8	2769.2272	2769.6050	32.76
**Conj4**	Ac-NleEHfRWGK(Dau=Aoa)-NH_2_	12.9	1694.7876	1694.7692	18.36

^1^ Retention time on Phenomenex Jupiter C12 column, gradient: 5–100% B, 20 min. According to the high-performance liquid chromatography (HPLC) analysis, the purity of the conjugates was always above 95%. ^2^ Mmo meas. (monoisotopic molecular mass) measured on a Thermo Scientific Q Exactive Focus Hybrid Quadrupole-Orbitrap mass spectrometer.

**Table 2 ijms-25-01095-t002:** In vitro antiproliferative activity of full-length α-MSH drug conjugates.

Cell Line	IC_50_ ^1^ (µM)				Relative Potency ^2^		
	Conj1	Conj2	Conj3	Dau	Conj1	Conj2	Conj3
B16	2.9 ± 0.6	2.8 ± 0.7	2.0 ± 0.7	0.026 ± 0.008	0.0090	0.0093	0.0130
A2058	9.8 ± 5.4	3.2 ± 0.4	3.0 ± 0.8	0.04 ± 0.007	0.0041	0.0125	0.0133
M24	12.8 ± 1.6	11.5 ± 0.4	11.0 ± 0.8	0.119 ± 0.025	0.0093	0.0103	0.0108
WM983B	9.9 ± 1.5	7.9 ± 0.7	3.6 ± 0.2	0.050 ± 0.023	0.0050	0.0063	0.0138

^1^ IC_50_ values (average ± SD) were determined by computerized curve-fitting program (GraphPad, La Jolla, CA, USA). Nonlinear regression analysis was performed with Prism 6 software to generate sigmoidal dose–response curves from which the 50% inhibitory concentration (IC_50_) values of the compounds were calculated and presented as micromolar (µM) units. Values shown are mean ± SD of two or three independent experiments, each performed in four parallels. The *p*-value of <0.01 was considered statistically significant. Statistical analysis was performed using Excel (version: 365; Microsoft, Redmond, WA, USA) employing the *t*-test. We estimate the average logIC_50_ (equivalent to the geometric mean of the IC_50_) over 3 curves as the weighted mean of the individual logIC_50_s. ^2^ Relative potency was calculated as the ratio of IC_50_ values of free Dau and conjugates.

**Table 3 ijms-25-01095-t003:** In vitro cytostatic effect of the Dau–α-MSH conjugates.

Code	IC_50_ (µM) ^1^					
	OCM-1	OCM-3	SK-MEL-202	WM983A	WM983B	A2058
**Free Dau**	0.73 ± 0.06	0.56 ± 0.05	0.045 ± 0.12	0.19 ± 0.06	0.33 ± 0.05	0.12 ± 0.07
**Conj2**	2.51 ± 0.06	2.39 ± 0.06	0.13 ± 0.09	0.79 ± 0.05	1.00 ± 0.00	0.95 ± 0.12
**Conj4**	29.68 ± 0.06	25.47 ± 0.08	2.06 ± 0.10	13.50 ± 0.06	10.18 ± 0.06	7.15 ± 0.07
**Targeting Index (TI) ^2^**						
**Conj2**	2.30	1.85	2.74	1.90	2.61	1.00
**Conj4**	1.47	1.31	1.30	0.84	1.93	1.00

^1^ IC_50_ values were determined by a computerized curve-fitting program (GraphPad Prism 6, nonlinear regression was estimated using the Levenberg–Marquardt method). Cells were treated with the compounds at 0.003–200 μM concentration range dissolved in the corresponding serum-free medium for 72 h. After the treatment, the PrestoBlue assay was carried out to determine the IC_50_ values. Values shown are mean ± SEM of three independent experiments, each performed in three parallels. ^2^ TI was determined using the equation of TI = ((IC_50_neg)/(IC_50_pos)conjugate)/((IC_50_neg)/(IC_50_pos)free drug)).

## Data Availability

Data is contained within the article.

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
