# Peer review of "Targeting the Melanocortin 1 Receptor in Melanoma: Biological Activity of α-MSH–Peptide Conjugates"

_ijms, 2024, doi:10.3390/ijms25021095_

Round 1

Reviewer 1 Report

Comments and Suggestions for Authors

In the manuscript by Hdiko Szabo and coworkers, the authors study different daunorubicin (daunomycin) and alpha-melanocyte-stimulating hormone conjugates efficacy in cell viability and tumor growth experiments. Concomitantly, the authors assess the correlation between the conjugate efficacy and cellular uptake, notably to the lysosomal and nuclear compartments, in the context of achieving increased drug efficacy by improving anti-neoplastic drug delivery to the cell nucleus (daunorubicin is a DNA minor groove binding entity, inhibiting the progression of topoisomerase II among other MOAs), and reduced toxicity by targeted delivery to melanocortin type 1 receptor-expressing cells. This is an enticing idea that others have explored before by suggesting or taking advantage of cell surface receptor trafficking (as reviewed by Kumar A et al. Front Pharmacol. 2020 27;11:629 and exemplified by Ziffert I et al. ChemMedChem. 2021 8;16(1):164-178). Unfortunately, several methodological, experimental, analysis, and interpretation omissions hold back the efforts put forward by the authors. These omissions are summarized below.

1 - The introduction detracts and does not support the aim of the study. Notably, the authors mention and describe the shortcomings related to the use of dacarbazine, a DNA alkylating drug used to treat several forms of melanoma, but later reveal, with no justification, that the study will be centered on daunorubicin alpha-MSH conjugates. As a curiosity, the latter is used as part of the chemotherapy cocktails to treat Hodgkins and non-Hodgkin lymphomas. Furthermore, in the introduction section, the authors offer an unwarranted discussion on melanocyte distribution, melanoma types, MC1R function, and genetic variants that worsen the dispersity. This reviewer strongly urges the authors to streamline the introduction section by staying on topic (i.e., targeted drug distribution for treating melanomas). An opening paragraph on the impact of this disease and the role of MC1R would suffice as an opening statement. In this context, avoid the use of nontechnical or cliche phrases such as: “Physiologically, more than 80 variants of MC1R have been recently described […] (cAMP) production as strongly as the wild-type receptor…”. In this example (many others are present, the word “Physiologically” has no bearing on the rest of the paragraph (why is it physiological? Variants with no apparent deleterious effect are not either physiological or non-physiological; they are just variants), and “strongly” is nontechnical verbose that should be replaced by pharmacological “potency.”

2 - Section 2.1 has synthesis details that are better relegated to the methods section. In fact, many of the elements mentioned here are repeated in the methods. 

On the topic of synthesis, it was surprising to see that the authors used a notoriously short-lived serum half-life peptide scaffold with no MC1R specificity. On this topic, no serum stability experiments were reported, a crucial omission that would invalidate any future development of this strategy. The authors should at least discuss their peptide scaffold use (if any reasonable justification could be found, this reviewer would be curious to find out…). Perhaps more concerning is the choice of a very short drug-peptide linker and the lack of direct evidence that the conjugates even bind or elicit cAMP responses from M1R or other melanocortin receptor subtypes, for that matter. The question surrounding ligand binding is a significant omission and shortcoming that the authors should address in a revised version of their manuscript. Following this vein, and in light of several melanocortin receptor-ligand structures reported in the last three years, this reviewer found the authors’ choice to conjugate daunorubicin to Lys11 surprising since these entities would likely not bind to the receptor with high affinity. To reiterate, any additional assertions or conclusions lack any support without functional or binding experiments to back to the authors’ strategy. This reviewer strongly urges the authors to perform functional or binding experiments with the different conjugates to demonstrate the interaction with MC1R or adequately justify why this is not warranted. The authors should also, at their discretion, characterize the serum half-life of the conjugates.

3 - A significant drawback is the need for more statistical rigor and proper data analysis. If the authors had reported the IC50 lethality doses as -Log values, converting geometric means into arithmetic, they would have realized that a 2 to 5-fold effect on IC50 is not significant (Table 2). This is at the root of the lack of correspondence between cell viability, cell penetration, and in vivo results. The authors were misguided by not confirming the possibility that the null hypothesis was true. This is the most salient shortcoming of the manuscript and the root of conflicting interpretations offered in the discussion and conclusion sections. To reinforce this, the authors do not provide any variance testing or do not show it in any part of the results. This reviewer cannot sufficiently stress the importance of hypothesis testing to draw conclusions from a data set. P values are not mentioned in any part of the manuscript. Additionally, this reviewer did not find any instance where the t-test or the Mann-Whitney test could be correctly applied. This reviewer strongly urges the authors to seek the advice of an accredited biostatistician to validate the results in a revised manuscript. 

4. The description of controls used in experiments is minimal. For example, what housekeeping amplicons (if any)were used in qPCR experiments? Was the (-)delta-delta-CT method used to show fold differences in MC1R expression? Do these results correlate with transfection experiments to validate the PCR results? How do mRNA levels correlate with cell surface receptor expression?

5 - Other not-so-minor observations… when dosing animals, a dose of 12.5 versus 25 is essentially the same dose. Dimensionwise, 10-fold differences in dosage would be adequate when assessing the effect of different amounts. The same applies to tumor inoculates. 5x10^5 is essentially the same as 1x10^6. Figure 6 is a clear testament to this. 

Comments on the Quality of English Language

The introduction section lacks coherence and is not relevant to the main aim of the manuscript. The results section is heavy on methods description that should be relegated to the "methods" section. The "Conclusions" section should follow the discussion. The discussion should include a contrast of the results with previously published data. Avoid the use of nontechnical or nonstandard verbose. For example, it is not appropriate to say that the "literature allows for the replacement of Mte bt Nle. Instead, provide a technical justification.

Author Response

Detailed answer to reviewer 1:

We thank the referee for their important comments on the abstract, the introduction as well as the results. We have modified the manuscript based on the suggestions.

Answers and comments for Reviewer:

1 - The introduction detracts and does not support the aim of the study. Notably, the authors mention and describe the shortcomings related to the use of dacarbazine, a DNA alkylating drug used to treat several forms of melanoma, but later reveal, with no justification, that the study will be centered on daunorubicin alpha-MSH conjugates. As a curiosity, the latter is used as part of the chemotherapy cocktails to treat Hodgkins and non-Hodgkin lymphomas. Furthermore, in the introduction section, the authors offer an unwarranted discussion on melanocyte distribution, melanoma types, MC1R function, and genetic variants that worsen the dispersity. This reviewer strongly urges the authors to streamline the introduction section by staying on topic (i.e., targeted drug distribution for treating melanomas). An opening paragraph on the impact of this disease and the role of MC1R would suffice as an opening statement. In this context, avoid the use of nontechnical or cliche phrases such as: “Physiologically, more than 80 variants of MC1R have been recently described […] (cAMP) production as strongly as the wild-type receptor…”. In this example (many others are present, the word “Physiologically” has no bearing on the rest of the paragraph (why is it physiological? Variants with no apparent deleterious effect are not either physiological or non-physiological; they are just variants), and “strongly” is nontechnical verbose that should be replaced by pharmacological “potency.”

 The reviewer’s has right that the functional description of drugs used for melanoma does not support the aim of the study. Therefore, this part was shortened and mentioned that they are not effective enough, and new drug development is necessary to improve the efficacy of the treatment. We also explained more precisely why peptide – daunomycin conjugates are good tool in development of small molecule drug conjugates for targeted tumor therapy and not only against Hodgkin and non-Hodgkin lymphomas. The description of the function of MC1R is also too long these additional effects might not adequate in targeted therapy, where the main function of the receptor is to promote the receptor mediated drug uptake by cells. We shortened this paragraph and explained it function in targeted therapy. We also changed the not appropriate word physiologically, etc.) in accordance to the reviewer’s suggestions.

2 - Section 2.1 has synthesis details that are better relegated to the methods section. In fact, many of the elements mentioned here are repeated in the methods. 

Thank you for your suggestion. The reviewer is right, we have modified the text to focus on avoiding repetition and on the actual results.

On the topic of synthesis, it was surprising to see that the authors used a notoriously short-lived serum half-life peptide scaffold with no MC1R specificity. On this topic, no serum stability experiments were reported, a crucial omission that would invalidate any future development of this strategy. The authors should at least discuss their peptide scaffold use (if any reasonable justification could be found, this reviewer would be curious to find out…). Perhaps more concerning is the choice of a very short drug-peptide linker and the lack of direct evidence that the conjugates even bind or elicit cAMP responses from M1R or other melanocortin receptor subtypes, for that matter. The question surrounding ligand binding is a significant omission and shortcoming that the authors should address in a revised version of their manuscript. Following this vein, and in light of several melanocortin receptor-ligand structures reported in the last three years, this reviewer found the authors’ choice to conjugate daunorubicin to Lys11 surprising since these entities would likely not bind to the receptor with high affinity. To reiterate, any additional assertions or conclusions lack any support without functional or binding experiments to back to the authors’ strategy. This reviewer strongly urges the authors to perform functional or binding experiments with the different conjugates to demonstrate the interaction with MC1R or adequately justify why this is not warranted. The authors should also, at their discretion, characterize the serum half-life of the conjugates.

The reviewer’s has right that the blood serum/plasma stability of a potential drug candidate is very important.  The manuscript describes a first step development of α-MSH – drug conjugates for melanoma targeting to find the potential conjugation site and length of appropriate homing peptide. During the in vitro experiments no or very tiny serum content is used in the applied circumstances (cellular uptake and localization studies were performed in incomplete medium) However the determination of cytostatic effect of the conjugates was performed in 2.5% FBS containing cell culture medium, after the 24h incubation the cells were washed and serum containing fresh medium is used for further incubation. However, in this case only the bound and internalized α-MSH – Dau are present, therefore the low stability has no influence on the cellular uptake and cytostatic effect. In in vivo experiments mice were treated i.p. that adsorb directly by s.c. developed tumors or close to tumors therefore, the conjugates are not exposed for long time to the blood serum or plasma. The tumor growth inhibition was detected and differences were observed between the conjugates. The results suggest the promising conjugation site that can be the base of further structural optimization. It is expected that based on these results further development of more potential conjugates with higher enzyme stability will lead to potential drug candidates.

3 - A significant drawback is the need for more statistical rigor and proper data analysis. If the authors had reported the IC50 lethality doses as -Log values, converting geometric means into arithmetic, they would have realized that a 2 to 5-fold effect on IC50 is not significant (Table 2). This is at the root of the lack of correspondence between cell viability, cell penetration, and in vivo results. The authors were misguided by not confirming the possibility that the null hypothesis was true. This is the most salient shortcoming of the manuscript and the root of conflicting interpretations offered in the discussion and conclusion sections. To reinforce this, the authors do not provide any variance testing or do not show it in any part of the results. This reviewer cannot sufficiently stress the importance of hypothesis testing to draw conclusions from a data set. P values are not mentioned in any part of the manuscript. Additionally, this reviewer did not find any instance where the t-test or the Mann-Whitney test could be correctly applied. This reviewer strongly urges the authors to seek the advice of an accredited biostatistician to validate the results in a revised manuscript

Our working hypothesis was based on the fact that the role of the peptide carriers is to make the Dau compound selective and target cell-specific. This property of the carrier peptides has been demonstrated in several of our previous publications (also included in the Introduction section). We emphasize the specificity of the carriers throughout this paper. Therefore, as a proof of concept, we have studied fundamental structure-activity relations in this work, and the earliest stage decisions about Dau and Dau-carrier peptide conjugates were based on their in vitro activity. Each Dau-conjugate was compared to the Dau in the overall study, and first, the activity was enumerated as the IC50 value (the compound concentration required for 50% cytotoxic or cytostatic activity). In practice, in the case of all compounds, three separate experiments were carried out, and each compound’s IC50 endpoint was determined by fitting a sigmoid function to measurements (responses) at a range of treatments (doses), constituting the dose-response curves. When the sigmoid curve can be provided to the experimental points, and the fitted IC50 is within the range of doses used, a point value is determined (e.g., IC50 = 40 nM = 0.04 µM, etc.). We calculated the average IC50 (equivalent to the geometric mean of the IC50) from three curves as the weighted mean of the individual IC50 values; our optimized viability assays showed that all compounds behaved reproducibly (with relatively small SD values). Therefore, the 2-5-10-fold differences in IC50 values are significant. The reproducibility of the activity of the standard compound (in our case, compound Dau) is usually also considered a good indication of the stability of an assay. The IC50 values of the compounds were compared to the IC50 value of the standard Dau compound. Based on these values, the effect of each Dau-peptide conjugate differed significantly from the effect of Dau (p <0.01 on each cell type). This was not explicitly indicated in the Tables but has now been corrected and is shown in the Table footnote. The p-value of <0.01 was considered statistically significant. Statistical analysis was performed using Excel (version: 365; Microsoft, Redmond, WA, USA) employing the t-test. The IC50 value of the compounds themselves does not determine the in vivo effect, penetration ability, or bioavailability. It provided us a guide to the impact of a particular Dau-conjugate in a specific type of cell. It was the first screening for the ranking of Dau-conjugates to be selected for subsequent in vivo experiments. The main message of the work, and we believe this can be followed in the evaluation of the results and the Conclusion section, is that Dau administered with peptide carriers (as peptide conjugates) can significantly achieve tumor-specific effects on target cells, improving the quality of life of the treated animals.

As we wrote in the method section (line 668-671) in the case of the in vivo allograft model, statistical analyses were performed by GraphPad Prism 6 (GraphPad Software, San Diego, CA, USA) using the non-parametric Mann–Whitney test, where p-values lower or equal than 0.05 were considered statistically significant. The revised manuscript include an own issue to the Statistical analysis (5.13; line 667-671)

  1. The description of controls used in experiments is minimal. For example, what housekeeping amplicons (if any) were used in qPCR experiments? Was the (-)delta-delta-CT method used to show fold differences in MC1R expression? Do these results correlate with transfection experiments to validate the PCR results? How do mRNA levels correlate with cell surface receptor expression?

We modified the text to include the housekeeping gene (lines 536-538): Relative expression was determined by normalizing the expression levels to human β-actin mRNA levels as housekeeping genes and by comparing the MC1R mRNA levels of the investigated cell lines to the healthy skin fibroblast cell line, CCD-986-Sk. To estimate the relative expression, the delta-delta Ct method was used. The data represents three independent experiments, each performed in 3 technical replicates.

It has been reported that MC1R antagonists are able to reduce MC1R protein levels without interfering with the mRNA expression (PMID: 14500544), however, in other articles it was also shown that mRNA levels of MC1R is correlates with protein expression (PMID:16420249). According to these results, MC1R mRNA expression may be leaky and cells which do not express detectable levels of MC1R protein do express detectable (although low) levels of MC1R mRNA. On the other hand, high levels of MC1R mRNA suggest high levels of MC1R protein expression. This article also reported data on transfecting cells with MC1R, showing that mRNA levels are high in transfected cell lines, however, this does not result in surface expression of the MC1R protein. In our experiments, we also showed that cell lines which seem to be more sensitive to PDCs might not be the ones which express the highest levels of mRNA. However, our results strongly support that cell lines which showed low IC50 values compared to the others, exhibit higher uptake of the peptide-drug conjugates too, suggesting selectivity and differential expression of the MC1R protein.

5 - Other not-so-minor observations… when dosing animals, a dose of 12.5 versus 25 is essentially the same dose. Dimension wise, 10-fold differences in dosage would be adequate when assessing the effect of different amounts. The same applies to tumor inoculates. 5x10^5 is essentially the same as 1x10^6. Figure 6 is a clear testament to this. 

According to our experience and according to the literature, using a 10-fold difference in concentration in the case of daunomycin conjugates would not be sufficient to determine the concentration of the peptide-drug conjugates accurately that should be applied for tumor growth inhibition experiments. In the case of Conj2, we performed a toxicity assay, distinctly demonstrating that a concentration of 5 mg/kg does not seem to be toxic and results in no crucial weight loss of mice after 4 treatments in 22 days. Conversely, a concentration of 10 mg/kg applied of the same conjugate only two times leads to a weight loss of approx. 20% which is generally considered to be a cut-off value and mice should be terminated upon reaching it.

When determining the amount of cells to use for the establishment of murine experimental model, we clearly showed in Figure 6 that there are cell lines with great tumor establishment capabilities (such as OCM-1) which are able to initiate tumor growth regardless of the cell concentration used. However, in the cases of OCM-3 and SK-MEL-202, there is an explicit difference when applying the cells in concentrations 500 000/mice, 1 000 000/mice and 2 000 000/mice. As an example, 500 000 cells/mice results in approx. a tumor size of 300 mm3 on Day40, in contrast, in the case of 1 000 000 cells/ml, the tumors are approx. 900 mm3 in size. We also have to note that the application of too many cells may cause exhaustion of the tumor cells due to starvation. Furthermore, since 100 µl of cell suspension was applied, cells were inoculated in a concentration of 10 000 000 cells/ml which is the highest concentration that is generally considered good lab practice. A cell suspension with 100 000 000 cells/ml would be rather challenging to handle and is not recommended. Since animal handling guidelines suggests no higher volume injected subcutaneously to mice than 200 µl, the concentration of 10 000 000 cells/mice would not be in accordance with good lab practice in any way.

The introduction section lacks coherence and is not relevant to the main aim of the manuscript. The results section is heavy on methods description that should be relegated to the "methods" section. The "Conclusions" section should follow the discussion. The discussion should include a contrast of the results with previously published data. Avoid the use of nontechnical or nonstandard verbose. For example, it is not appropriate to say that the "literature allows for the replacement of Mte bt Nle. Instead, provide a technical justification.

We modified the discussion section based on the instructions of Referee.

Reviewer 2 Report

Comments and Suggestions for Authors

Targeted toxins are becoming increasingly important in advanced tumor therapies. The authors investigated targeting and efficacy of conjugates comprising alpha-melanocyte-stimulating hormone and daunomycin. This is important because advanced treatments for diseases such as metastatic malignant melanoma are urgently needed. The authors systematically investigate four different conjugates differing in position and number of daunomycin in vitro and in vivo. The experimental results are clearly described but the manuscript lacks a clear structure and elaboration of the novelties. The strategy for selecting the experiments to be conducted was not convincingly written. Overall, the results provide clear indications that some conjugates should be preferred over others, but the underlying structure-function relationship remains unclear and does not allow rational conclusions to be drawn for future developments.

Specific comments:

1.     Abstract: The abstract is very unspecific since it only contains qualitative statements (“still rather low”, “differential expression”, “lower expression”, great antiproliferative effect”, “inhibit tumor expansion significantly”). The authors should provide specific numbers for all statements.

2.     Abstract, line 23: The abbreviation alpha-MSH must be introduced.

3.    Abstract, lines 24–25: The authors do not explain why they designed and synthesized MSH analogs, which is important for understanding the rationale of the study.

4.     Abstract, lines 26–30: The authors described what they did, but they should rather describe why they did it.

5.     Introduction, lines 60–77: Instead of the historical overview, the authors should instead focus on newest developments in NOTA-conjugated lactam-cyclized MSHs and other new types of MSH conjugates. 

6.     Introduction: The authors correctly described the developments of MSH analogs and conjugates, but they do not elaborate the novelties of their study. MSH-daunomycin conjugates were already described 46 years ago in 1977 (the work was cited by the authors). On the basis of this development the authors should clearly state what is different in their work compared to previous publications since then.

7.     Introduction, line 124 and Discussion, line 407: The authors do not explain their motives for choosing oxime ligation in the introduction. In the discussion, they say the bond is non-cleavable but in the next sentence they explain release in lysosomes. The authors should better explain their motives for the use of oxime linkage and provide more specific data from the literature, e.g. on release rate in lysosomes.

8.     Results, line 132: For better understanding, the authors should explain that -NH2 indicates a C-terminal amide modification (and not an amino group).

9.     Results, line 144: The authors should introduce the abbreviation Aoa here for later use.

10.     Results, Table 2: The authors should use the same units for conjugates and Daunomycin to avoid confusion with the numbers.

11.     Results, Table 2 and Table 3: The authors should provide information, which algorithm has been used for curve-fitting, e.g. least squares method in a 4-parameter fit.

12.     Results, line 226: The measurement of body weight is not sufficient to measure acute toxicity. Daunomycin conjugates could result in liver, kidney, and heart damage. The authors should at least provide blood parameters for liver, kidney or heart (ALT, AST, creatinine, ANP) and/or histologic analyses of the tissues.

13.     Results. The authors should explain why they changed from MTT-assay (Table 2) to PrestoBlue assay (Table 3). The IC50 of Conj2 for A2058 is 3.2 µM in Table 2 and 0.95 µM in Table 3, and for WM983B 7.9 µM in Table 2 and 1.00 µM in Table 3, which makes a comparison of Conj4 to the other conjugates difficult.

14.     Results, line 389: The authors write “the tumor size … was reduced to 55% of the control size” but the tumor size was not reduced at all but was still growing, so the correct wording is that “the growth of the tumor … was reduced to 55% of the control size”.

15.     Results, line 391 and Discussion, line 457: Provide the p-value for significance.

16.     Methods, line 623: How were the groups created? Completely random or was this a stratified procedure, e.g., to obtain groups with the most equal mean and standard deviation of tumor sizes between groups? Please explain how groups were established.

17.     Discussion: The discussion is rather weak and should include and compare recent results on MSH conjugates.  Underlying structure-function relationships on receptor binding and internalization should be discussed. The current discussion is rather a repetition of the results.

18.     Supplement, Table S2: Tumor weight is tumor weight in %?

Comments on the Quality of English Language

Minor editing of English language required.

Author Response

Detailed answer to Reviewer 2:

We thank the referee for their important comments on the abstract, the introduction as well as the results. We have modified the manuscript based on the suggestions.

Answers for Reviewer:

  1. Abstract: The abstract is very unspecific since it only contains qualitative statements (“still rather low”, “differential expression”, “lower expression”, great antiproliferative effect”, “inhibit tumor expansion significantly”). The authors should provide specific numbers for all statements.

We modified the abstract according to the suggestions of referees.

  1. Abstract, line 23: The abbreviation alpha-MSH must be introduced.

We introduced the abbreviation of alpha-MSH in Line 25

  1. Abstract, lines 24–25: The authors do not explain why they designed and synthesized MSH analogs, which is important for understanding the rationale of the study.

We have explained the reason of preparation and biological evaluation of α-MSH conjugates (lines 23-26)

  1. Abstract, lines 26–30: The authors described what they did, but they should rather describe why they did it.

We rephrased the abstract to explain why these α-MSH conjugates have been designed, prepared, and tested. 

  1. Introduction, lines 60–77: Instead of the historical overview, the authors should instead focus on newest developments in NOTA-conjugated lactam-cyclized MSHs and other new types of MSH conjugates. 

Thanks for the referee’s suggestions. According to this observation, we have modified the introduction to focus on newly designed and applied α-MSH derivatives. 

  1. Introduction: The authors correctly described the developments of MSH analogs and conjugates, but they do not elaborate the novelties of their study. MSH-daunomycin conjugates were already described 46 years ago in 1977 (the work was cited by the authors). On the basis of this development the authors should clearly state what is different in their work compared to previous publications since then.

As requested by the reviewer, we have explained the differences between the previously published and the now presented conjugates (lines 101-102).

  1. Introduction, line 124 and Discussion, line 407: The authors do not explain their motives for choosing oxime ligation in the introduction. In the discussion, they say the bond is non-cleavable but in the next sentence they explain release in lysosomes. The authors should better explain their motives for the use of oxime linkage and provide more specific data from the literature, e.g. on release rate in lysosomes.

The reviewer is right that we did not explain why we chose oxime bonding when designing our conjugates. This omission has been corrected (lines 109-117).

  1. Results, line 132: For better understanding, the authors should explain that -NH2 indicates a C-terminal amide modification (and not an amino group). Thank you for this observation. We have filled the omission and we explained the -NH2 as follows: N-terminal acetylated and C-terminal amidated native tridecapeptide (lines 82-83 as well as line 127)

  1. Results, line 144: The authors should introduce the abbreviation Aoa here for later use.

 In line 140 we have introduced the Aoa abbreviation.

  1. Results, Table 2: The authors should use the same units for conjugates and Daunomycin to avoid confusion with the numbers.

Thank you for your comment. We have uniformed the orders of magnitude of the IC50 values, in all cases µM is presented.

  1. Results, Table 2 and Table 3: The authors should provide information, which algorithm has been used for curve-fitting, e.g. least squares method in a 4-parameter fit.

The following text has been added to the footnote below the:

Table 2: Nonlinear regression analysis was performed with Prism 6 software (GraphPad, La Jolla, CA, USA) to generate sigmoidal dose-response curves from which the 50% inhibitory concentration (IC50) values of the compounds were calculated and presented as micromolar (µM) units.

Table 3: Nonlinear regression was estimated using the Levenberg-Marquardt method.

  1. Results, line 226: The measurement of body weight is not sufficient to measure acute toxicity. Daunomycin conjugates could result in liver, kidney, and heart damage. The authors should at least provide blood parameters for liver, kidney or heart (ALT, AST, creatinine, ANP) and/or histologic analyses of the tissues.

We agree with the reviewer that the description of the liver toxicity assessment should be more detailed. Histological samples of the liver were evaluated by a pathologist and liver histology from the free drug-treated and the other groups were compared. In their opinion, the difference was not significant. To support our findings on the non-toxicity of the liver in the conjugate-treated mice, we calculated the liver weight/body weight and the spleen weight/body weight ratios. The results clearly show that the liver weights of the free drug treated mice were significantly reduced compared to the body weights of the free drug treated mice compared to the other groups.

We understand and endorse the Reviewers’ point of view that toxicity testing of new compounds is essential for drug development process and must be addressed to achieve the full potential of new drug delivery system. The preclinical toxicity testing on various biological systems reveals the species-, organ- and dose-specific toxic effects of an investigational product. The toxicity of substances can be observed by in vitro studies using cells/cell lines and in vivo exposure on experimental animals (Parasuraman, Journal of pharmacology & pharmacotherapeutics, 2011, 2(2), 74-9.). Animal weight and behavioral changes are the critical tool in toxicity testing as animals should be protected from stress and pain (Council NR. Guide for the Care and Use of Laboratory Animals. National Academies Press; 2010.). Dau is known to be rapidly and widely distributed in tissues, whereby the highest levels were found in the liver, spleen, kidneys, lungs and heart (Danesi et al., Eur J Cancer Clin Oncol. 1988, 24 (7), 1123–1131.). Since the liver is the vital organ in metabolism of Dau, production of a toxic intermediates which may trigger liver injury and impair with the liver function increasing the risk of toxicity (Paul et al., Cancer Lett. 1980, 9 (4), 263–269.). Analysis of organ weight in toxicology studies is an important factor for identification of potentially harmful effects of drugs (Michael et al., Toxicol Pathol. 2007, 35 (5), 742–750.), thus the liver weight analysis provides an understanding of drug toxicity (Kuntzman et al., J. Pharmacol. Exp. Ther. 1966, 152 (1), 151–156.). To investigate toxic effects and the tolerance of the Dau conjugates we determined the animal weight, behavioral and appearance changes and liver weight of the animals. We neither detected signs of behavioral changes nor significant body or liver weight changes administrating the Dau conjugates, while significant changes have been obtained for the group treated with Dau. This may imply that the conjugates have a lower toxicity.

In toxicity evaluation of compounds we have a panel of parameters that we follow during the procedure such as general look of the animal, quality of fur, mobility, facial expression, body condition, presence of pain and tremor, behavior and body weight. The body weight changes are the most informative because a decrease of the body weight by 20% compared to the start of experiment tell us about high toxicity of the tested compound (PMID: 31665969).

We understand the reviewer point of view that changes in the liver are often reflected by biochemical abnormalities of liver function. To analyze the level of circulating aminotransferases can serve as markers of hepatocellular injury (McGill, EXCLI J. 2016, 15:817–828.) and provide useful information about the liver toxicity, but unfortunately, we did not evaluate circulating levels of aminotransferase in the present study as well as the liver histology because we focused liver toxicity determination on liver weight as described above.

  1. Results. The authors should explain why they changed from MTT-assay (Table 2) to PrestoBlue assay (Table 3). The IC50 of Conj2 for A2058 is 3.2 µM in Table 2 and 0.95 µM in Table 3, and for WM983B 7.9 µM in Table 2 and 1.00 µM in Table 3, which makes a comparison of Conj4 to the other conjugates difficult.

The main reason for the change was efficiency. Since MTT and PrestoBlue assays are interchangeable (PMID: 35659575) and PrestoBlue shows better specificity and efficacy (PMID: 35659575, 25464019, 24103906), we decided to switch our protocol for in vitro antiproliferative evaluation from MTT to PrestoBlue. Furthermore, according to their safety data sheet, MTT is able to cause genetic defects, whereas PrestoBlue is not. Another advantage is that PrestoBlue is a ready-made and easy-to-use solution, whereas MTT requires weighing and resolution, which can be quite error-prone.

A comparison of the two experiments was not performed in our manuscript. We used our first experiment to distinguish the first set of conjugates and to select the lead compound. Subsequently, we developed several conjugates using Conj2 as a starting point and only the newly synthesized conjugate, which was tested in an antiproliferative assay, was compared to Conj2 using PrestoBlue assay.

  1. Results, line 389: The authors write “the tumor size … was reduced to 55% of the control size” but the tumor size was not reduced at all but was still growing, so the correct wording is that “the growth of the tumor … was reduced to 55% of the control size”.

Thank you for your comment, it was indeed a typo. We have reworded.

  1. Results, line 391 and Discussion, line 457: Provide the p-value for significance.

Line 391: Tumor growth inhibition of Conj2 vs controls: p-value of 0.0056

Line 457: Significant weight loss of Dau vs controls: p-value of 0.0164. However, the main cause to terminate animals was not a statistically significant weight loss compared to control, rather the fact that mice treated with free Daunomycin reached a cut-off value of 20% of weight loss.

  1. Methods, line 623: How were the groups created? Completely random or was this a stratified procedure, e.g., to obtain groups with the most equal mean and standard deviation of tumor sizes between groups? Please explain how groups were established.

Mice were randomized in a stratified procedure considering two main factors. The primary factor was to obtain groups with the most equal tumor sizes, the secondary factor was to have groups in which the weight of mice are the most equal. this explanation has been added to this part of the Method (lines 629-633)

  1. Discussion: The discussion is rather weak and should include and compare recent results on MSH conjugates.  Underlying structure-function relationships on receptor binding and internalization should be discussed. The current discussion is rather a repetition of the results.

Thank you for your comment, we have indeed conceptualized the discussion in general terms. In the current version we have tried to reformulate the discussion according to the reviewer's point of view.

  1. Supplement, Table S2: Tumor weight is tumor weight in %?

Thank you for your comment, it was indeed a typo. We have corrected it in the Table S1.

Round 2

Reviewer 2 Report

Comments and Suggestions for Authors

The authors carefully addressed the reviewer’s comments. Two points were not sufficiently clarified, but, according to the authors, cannot be corrected experimentally retrospectively. At least for one item, the toxicity studies, the limitations of that study should be stated.

Specific comments:

1.     Toxicity studies. I can understand the situation and the difficulty or impossibility of adding more convincing data. Regarding the publication by Kuntzman et al. that liver weight analysis provides an understanding of drug toxicity, it is over 50 years old and was published at a time when such analyses were more difficult. Today, commercial laboratories can determine these parameters in an hour for only a few cents, so there is no reason not to do it. So, it's a shame it wasn't done. Adding the information on liver weight is fine, but the authors should also add information about the limitations of the toxicity study to the manuscript.

2.     Viability assays: Adding the sentence “has higher specificity and efficacy, instead of the previously used MTT assay” should be sufficient here. In future, the authors should take care to use identical tests within one and the same study.

3.    Table S1: The authors corrected the table S1 but introduced a new typo: “Tumor weigh in %t”. Please correct again.

Comments on the Quality of English Language

Minor editing of English language required. 

Author Response

We thank the reviewer the comments and suggestions. Unfortunately, the organs like livers were kept in formalin for months therefore they cannot be used for the suggested toxicity study. To do it we have to make another in vivo experiment. Therefore, we provided the liver weights in the text and in supplementary information in detail as toxicity data for conjugates and daunomycin. However, thanks a lot for the suggestion and we will use this information in our further studies. Maybe we will make also comparative studies of different experiments. In the manuscript we signed the new parts with yellow background. We also revised the typo in the table of the supplementary information.